# Factors Associated with Return Visits by Elders within 72 Hours of Discharge from the Emergency Department

**DOI:** 10.3390/healthcare11121726

**Published:** 2023-06-12

**Authors:** Li-Hsiang Wang, Hui-Ling Lee, Chun-Chih Lin, Chia-Ju Lan, Pei-Ting Huang, Chin-Yen Han

**Affiliations:** 1Department of Nursing, China Medical University Hsinchu Hospital, Hsinchu 302, Taiwan; n56546@mail.cmuhch.org.tw; 2Department of Nursing, Chang Gung University of Science and Technology, Taoyuan 333, Taiwan; lhling@mail.cgust.edu.tw (H.-L.L.); cclin01@mail.cgust.edu.tw (C.-C.L.); 3Department of Nursing, New Taipei Municipal TuCheng Hospital, Chang Gung Medical Foundation, New Taipei City 236, Taiwan; 4Department of Nursing, Taichung Armed Forces General Hospital, Taichung 411, Taiwan; ting-an@803.org.tw; 5Department of Nursing, National Taiwan University Hospital Hsin-Chu Branch, Hsinchu 302, Taiwan; g81424@hch.gov.tw

**Keywords:** return visit, elder, emergency department, triage risk screening tool

## Abstract

Elders have a higher rate of return visits to the emergency department (ED) than other patients. It is critical to understand the risk factors for return visits to the ED by elders. The aim of this study was to determine the factors associated with return visits to the ED by elders. This study retrospectively reviewed the hospital charts of elders who returned to the ED within 72 h after discharge from ED. The risk factors identified in the Triage Risk Screening Tool were applied in this study. Of the elders discharged from the ED, 8.64% made a return visit to the ED within 72 h. The highest revisit rate occurred within 24 h after discharge. Factors associated with return ED visits within 24 h by elders were difficulty walking and having discharge care needs. The factor associated with ED return visits within 24–48 h was polypharmacy. Difficulty walking, having discharge care needs, and hospitalization within the past 120 days were associated with return visits made within 48–72 h following discharge. Identifying the reasons for return visits to the ED and providing a continuous review of geriatric assessment and discharge planning could reduce unnecessary revisits.

## 1. Introduction

Return visits to the emergency department (ED) are a global indicator of the quality of emergency care [1]. The ED return visit rate among adults aged 65 years and above is higher than other populations [1]. Elders require more health care and medical assistance because physical deterioration, cognitive decline, multimorbidity, and atypical presentation make the management of elders’ care needs more complex and difficult [2,3]. Return visits to the ED are associated with adverse events [4,5], additional medical expenditures [6], and a 3–10% increase in mortality rate [7]. Return visits may also indicate premature discharge, increasing the risk of negative health outcomes, patient safety impacts, medical disputes [4,5], ED workloads, and societal financial burden [8]. It is crucial to identify the factors associated with return ED visits by elders to improve ED care and reduce the number of unnecessary return visits to the ED.

The definitions of an ED return visit ranges from 48 h to 90 days between visits, but commonly refers to return within 72 h of discharge. According to previous studies, ED return visit rates for elders range from 7–16% [9,10,11]. An ED return visit rate below 5% is often used to indicate the quality of care in Taiwan and other countries [1,12]. The management of elders in EDs is difficult due to physical deterioration, cognitive decline, multimorbidity, and atypical presentation [2], which increase the risk of return visits by elders to the ED. The increasing aging population has a significant impact on the demand for ED services and presents a global challenge to emergency care. In Taiwan, elders aged 65 and above comprised approximately 17.56% of the total population in 2022 [13]. Taiwanese elders are the main users of emergency care, accounting for 24.92–28.5% of all ED visits by adults [14,15] and 40.8% of ED medical expenses among adults in Taiwan [16]. Identifying the factors that predict elders’ return visits can increase recognition by ED staff of high-risk patients and reduce return visits by elders.

ED return visits by elders cause negative physical, psychological, and social impacts on patients and family caregivers [17,18]. Literature has shown that ED return visits are avoidable [1 Han]. To reduce the risk of return visits by elders, ED healthcare providers need to be aware of the risk factors for return visits in elders. Previous studies have explored the risk factors for ED return visits by elders [4,5,19,20] but have not identified clear risk factors for return visits within different amounts of time following discharge. Identifying the factors associated with return visits to the ED within different amounts of time following discharge will help prevent elders’ return visits. The purpose of this study was to determine the factors associated with return visits to the ED by elders within 72 h after discharge and to predict the risk factors in return visits within different amounts of time after discharge.

## 2. Materials and Methods

### 2.1. Study Design and Setting

This is a retrospective review study. The medical charts of elders who returned to the ED from January to December 2017 were analyzed. The participating hospital is a regional hospital in Taiwan, with approximately 75,000 to 80,000 patients presenting annually to the ED. The overall rate of return visits was higher in 2017 than in the previous year, and the audit process revealed that the number of elder return visits was a significant indicator. On that basis, a comprehensive analysis of elders’ return visits should help to improve the quality of emergency care.

### 2.2. Participants

For the purposes of this study, the index visit was defined as the two ED visits made by an elder who returned within 72 h of ED discharge. The inclusion criteria for the study were: (1) ≥65 years of age and (2) returned to the ED within 72 h after discharge. The exclusion criterion was more than one return visit within 72-h.

### 2.3. Risk-Screening Tool

The risk factors identified by the Triage Risk Screening Tool (TRST) were applied in the study. The TRST is a simplified version of the Identification of Seniors at Risk (ISAR) measurement tool, developed to identify risks and predict ED return visit rates by elders. It was first modified by Meldon et al. [21] to stratify the risk of ED return visits and hospitalization in the elders [22]. The TRST assesses the following factors: history or evidence of cognitive impairment, difficulty walking, recent falls, polypharmacy (use of ≥5 medications), a visit to the ED in the previous 30 days, hospitalization in the past 120 days, whether the patient lives alone with no family caregiver, and professional evaluation by ED healthcare professionals. The presence of risk factors listed in the TRST was assessed by answers of ‘yes’ or ‘no’, and in cases where two ‘yes’ answers were recorded, the patient was deemed to be at high risk for an ED return visit. The area under the receiver operating characteristic (ROC) curve for TRST in predicting ED return visits and hospitalization within 30 and 120 days was 0.72 and 0.65, respectively. This indicates that the performance of the TRST was acceptable.

### 2.4. Data Collection

Using convenience sampling, participants were selected from the electronic medical records database at the participating ED. The 72-h return visit is one of the key audit indicators. The electronic record system ensures content integrity by including reminders to healthcare professionals to complete all relevant documentation; this helped to ensure that all eligible participants were included in the present study. This study initially set up all adults’ (≥18 years old) return visits covering the period from 1 January 2017 to 13 December 2017, and then recruited the elders’ (≥65 years old) return visits. As described below, a number of strategies were deployed to further ensure the validity and reliability of the study data.

When performing a retrospective review study, the investigators play a crucial role in ensuring data quality and must be trained accordingly [23]. To ensure the accuracy and consistency of data collection, all members of the research team attended meetings to confirm the relevant variables, procedures, and tools before commencing data collection. The Excel data collection tool was based on the Triage Risk Screen Tool. Two investigators (CJL and PTH) independently extracted the data of 30 participants. Any disagreements were resolved by discussion with a third investigator (LHW), who was responsible for analyzing the data. In total, 15 elders were found to have made more than one return visit within 72-h (see Figure 1 below). To protect patient privacy, all records and documentation collected during the study were anonymized and were not used for any other purpose. A chart review proposal was submitted to the participating hospital’s research ethics committee.

### 2.5. Data Analysis

After estimating the required sample size, the study data were assessed for normal distribution. Data were analyzed using IBM SPSS 20.0. The continuous variables are presented as the mean and standard deviation, and the categorical variables are presented as percentages. For subgroup analysis, data were stratified by age (65–74, 75–84 or ≥85 years), return visit time (within 24, 24–48, or 48–72 h after discharge), and triage level (1 and 2, emergent, ≥3, urgent). Differences in demographic and medical data were analyzed using the chi-square test. The potential predictors of return visits were tested by logistic regression. The association between risk factors and return visits after different amounts of time is expressed as the odds ratio (OR) and 95% confidence interval (CI). A *p* value < 0.05 was considered statistically significant in all tests.

## 3. Results

### 3.1. Characteristics of Return Visits by Elders

The total number of participants in the study was 1187, representing 27.17% of all ED visits. A total of 4.71% of all adult patients made return visits to the ED within 72 h following discharge, while 8.64% of all elders discharged from the ED returned within 72 h. The rate of return visits by male and female elders was similar. The majority of return visits were made by patients aged 65–74 (39.46%) and 75–84 years of age (38.62%), non-trauma patients, and patients at level three of triage (urgent) on the Taiwan Triage and Acuity Scale, and most return visits were made within 24 h (Table 1).

The primary reasons for a return visit within 72 h were new medical conditions (51.8%) or conditions pertaining to a previous illness or treatment (48.2%). The destination after return visits included discharge from the ED (66.3%) and hospitalization (29.2%). The top three TRST risk factors were a history of chronic disease (93.9%), living alone without a family caregiver (69.5%), and a previous ED visit in the past 30 days (49.9%) (Table 1). In a total of 24.5% of return visits, the patient was assessed by ED nurses as being in need of discharge care. The top two needs in discharge care were tubing care (49.0%) and wound care (24.5%) (Figure 2).

### 3.2. Times and Reasons for Return Visits by Elders

After analyzing the time characteristics of the attributes of return visits within 72 h, the results demonstrated that the return visit rate gradually decreased with time at different ages. The highest portion was a return visit within 24 h after discharge from ED. This accounted for 44.4% of return visits in aged 65–74 years, 42.1% of those aged 75–84 years, and 37.9% of those aged ≥85 years. There was no statistical significance among the age subgroups and return visit time, as well as no differences in gender. Analysis of follow-up outcomes of the ED return visit showed that the proportion of patients who ended up hospitalized decreased with time (Table 2).

For elders who made return visits due to conditions pertaining to a previous disease or treatment, a total of 53.8% of return visits were made within 24 h after discharge, 29.5% of return visits were made within 24–48 h, and 16.5% of return visits were made within 48–72 h. For elders who made return visits due to new medical conditions, the distribution of the three time points were similar. The results of an analysis of the reason and amount of time before the return visit demonstrated that the number of patients with conditions pertaining to a previous illness or treatment who returned to the ED within 24 h of discharge was higher than the number of patients who returned within 24 h of discharge with new medical conditions (OR, 2.57; 95% CI, 2.06–3.21; *p* < 0.0001). Comparing these two groups of elders with those who returned for different reasons, the patients who had conditions pertaining to a previous problem were less likely to return to the ED within 24–48 h of discharge (OR, 0.68; 95% CI, 0.54–0.85; *p* = 0.0008) or within 48–72 h after discharge (OR, 0.45; 95% CI, 0.35–0.58; *p* < 0.0001). An analysis of the destination of patients following return visits showed that the proportion of patients who ended up hospitalized decreased with the amount of time that lapsed between discharge and the return visit (Table 2).

### 3.3. Factors Associated with Return Visits by Elders

The results of an analysis of the distribution of TRST risk factors in elders who made return ED visits according to time point showed that almost half of elders with cognitive impairment returned within 24 h after discharge from the ED. Elders with cognitive impairment had a lower rate of return visits within 24–48 h after discharge (OR, 0.62; 95% CI, 0.43–0.91; *p* = 0.013) compared with those without cognitive impairment. The proportion of elders who were deemed to have care needs after discharge and who made return visits differed significantly among the time points, with 40.4% making a return visit within 24 h, 30.2% making a return visit within 24–48 h, and 22.3% making a return visit within 48–72 h. The elders who had care needs post-discharge had a significantly higher rate of return within 24 h after discharge (OR, 1.33; 95% CI, 1.02–1.74; *p* = 0.035). Elders with difficulty walking, polypharmacy, a return visit to the ED in the previous 30 days, hospitalization in the previous 120 days, and who were living alone without a family caregiver showed no statistically significant difference between the two time points (Table 2).

### 3.4. Distribution of Risk Factors for Return Visits by Elders

The TRST risk factors were further investigated by multivariate regression analysis. Only six patients in this study lived alone. This factor was excluded from the multivariate analysis. The results showed that difficulty walking and care needs after discharge were statistically associated with a risk of a return visit to the ED within 24 h. Elders with difficulty walking had a significantly increased risk of a return visit to the ED within 24 h (adjusted OR [aOR], 1.76; 95% CI, 1.23–2.50; *p* = 0.002) compared to those without difficulty walking. Elders who had care needs after discharge had a decreased risk of returning within the first 24 h (aOR, 0.66; 95% CI, 0.48–0.91; *p* = 0.012) compared to those who did not have care needs. The presence of cognitive impairment (aOR, 1.7; 95% CI, 1.06–2.72; *p* = 0.027) and polypharmacy (aOR, 1.4; 95% CI, 1.07–1.83; *p* = 0.015) significantly increased the risk of a return visit to the ED within 24–48 h after discharge. Multivariate analysis showed that the TRST factors significantly associated with a return visit to the ED within 48–72 h after discharge included difficulty walking, hospitalization during the previous 120 days, and the need for nursing care post-discharge. Elders with difficulty walking showed a decreased risk of making a return visit to the ED within 48–72 h after discharge (aOR, 0.57, 95% CI, 0.39–0.82, *p* = 0.003) compared to those without difficulty walking. Patients who had been hospitalized in the previous 120 days had an increased risk of making a return visit within 48–72 h after discharge (aOR, 1.52; 95% CI, 1.07–2.16; *p* = 0.019) compared to those without such a history. Elders assessed as needing nursing care post-discharge had an increased risk of making a return visit within 48–72 h (aOR, 1.62; 95% CI, 1.1–2.37; *p* = 0.014) compared to those without this need (Table 3).

## 4. Discussion

This retrospective review study analyses the factors associated with ED return visits by elders in Taiwan. Difficulty walking, post-discharge care needs, cognitive impairment, polypharmacy, and hospitalization in the previous 120 days were associated with return visits to the ED by elders. Elders in whom these factors were present were at greater risk of making a return visit to the ED within 72 h of discharge. The findings of this study are consistent with those of previous studies showing that difficulty walking and falls [24 COX], cognitive impairment [4], polypharmacy [4,5], and post-discharge care needs [5,20 Rising] led to increased ED return visits. Recent hospitalization is also a predictor of return visits. Previous studies have emphasized that elders with a history of a recent ED return visit [8] and those who have been hospitalized in the previous six months [24] should be managed more cautiously in the ED. Elders are subject to deterioration due to functional decline, cognitive impairment, frailty, malnutrition, and polypharmacy [25]. Geriatric assessment and recognition of the risk factors affecting elders are critical in the ED.

The present study showed that approximately one quarter of the elders returning to the ED had tubing and wound care issues after discharge; 70% and 80% of these patients, respectively, were discharged from the ED again. The findings of this study highlight the shortcomings of effective discharge planning related to tubing and wound care for elders. Similarly, the present study demonstrates that 66.3% of the elders returning to the ED were discharged again after evaluation. According to the literature, effective discharge planning is an important element in ensuring continuity of care for elders following discharge from the ED [1]. ED nurses play a key role in effective discharge planning, which should be individualized to meet the needs of elders [20]. Effective discharge planning helps elders and family members to deal with health problems at home [20]. The present findings suggest that ED nurses need to pay greater attention to elders with tubing and wound care issues. Elders’ self-care capabilities should also be evaluated before discharge to avoid unnecessary return visits. To ensure that the information in ED discharge planning is comprehensible for elders experiencing functional decline and multimorbidity, more thought should be given to content, method of instruction, and appropriate figures and font sizes.

Previous studies have found that 8.2–48% of ED return visits were avoidable [15,18]. As mentioned earlier, an ED return visit rate of over 5% indicates a need for improvement in the quality of ED care. A recent study showed that elders who made return visits to the ED had a lower rate of admission to the ICU and lower costs compared with patients who were directly admitted [8]. The study recommended that ED return visits should not be seen as a reflection of poor care [8]. However, elders comprise a vulnerable population with multimorbidity and complex health needs [20]. Non-urgent problems related to pre-existing health problems and chronic illnesses may be overlooked by ED healthcare providers [1]. ED healthcare providers need to have the knowledge to integrate geriatric assessment and management and to holistically assess patients’ health status, which may be complicated by both chronic and acute illnesses, as well as to provide adequate discharge planning.

The results of this study showed that 8.64% of elders who were ≥65 years of age made ED return visits within 72 h. The return visit rate in the present study was higher than that reported in studies conducted in other countries [26,27]. This indicates that the elders in Taiwan have a higher ED utilization. The return visit rate within 72 h of discharge among elders was also higher than the 5.6–7.2% reported in other studies [28]. In the present study, age was not significantly associated with return visit time, which is consistent with the results reported by Chen et al. [29] and Oiveira et al. [10] but contrasts with the findings of other studies showing that the return visit rate decreased with age in geriatric patients [4,5]. In this study, the return visit rate for elders aged ≥85 years was approximately 20% and the majority of elders aged ≥85 years were hospitalized after an ED visit. The present study also shows that the return visit rate was not significantly associated with gender in elderly patients. While our finding is consistent with some published studies [29], contrary results show that male patients had a higher return visit rate [4,5,28]. Regarding triage levels, Lowthian et al. [30] demonstrated that elders who were categorized as non-urgent had a higher return visit rate. In this study, we did find a significant difference in return visit rates between triage levels. However, 80% of the elders were triaged as non-urgent during their first visit to ED. The elders were not triaged again upon the ED return visit in Taiwan. Mobility is among the risk factors for elder return visits to ED [26]. The present findings indicate that elders with walking difficulties are at higher risk of return to the ED within 24 h. One possible explanation is that elders with walking difficulties still have unresolved medical problems following discharge from ED. Mobility issues may add to home care difficulties for the patient or their family members, and this group of elders may need to return to the ED for care within a short time.

One limitation of this retrospective review study is that it applied medical records from a single hospital, which may limit the generalizability of the research findings. Another limitation is that return visits to the ED by elders were not triaged again in Taiwan. Any change in medical status relating to a previous condition could not be assessed.

## 5. Conclusions

The results of this study may help to reduce ED return visits by elders. Based on the findings of this study, it is recommended that hospital administrators strengthen the ability of ED staff to perform comprehensive geriatric assessments. ED in-service training related to geriatric care should be regularly provided to ED staff to enhance their ability to recognize and assess high-risk elders. ED care should include effective discharge planning for elders with post-discharge needs. Further study should take into consideration the associated factors, and ED geriatric care models should be reviewed with a view to reducing elders’ return visits to ED. Identifying the relevant risk factors will help to prevent unnecessary return visits and will improve both quality of care and health outcomes for these patients.

## Figures and Tables

**Figure 1 healthcare-11-01726-f001:**
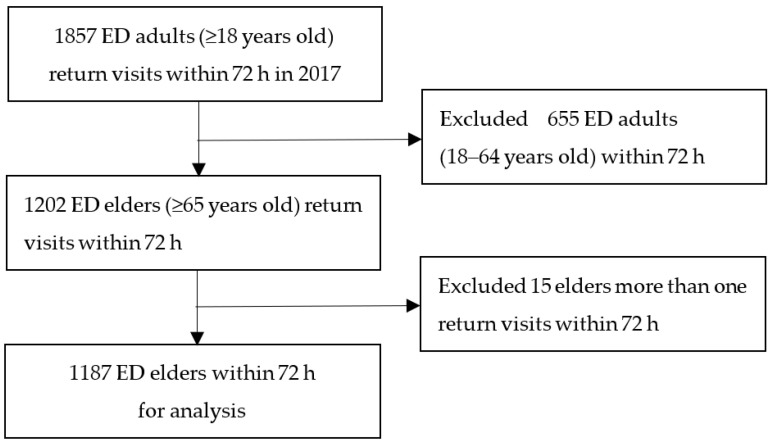
Flow chart for elders’ return visits within 72 h to the ED.

**Figure 2 healthcare-11-01726-f002:**
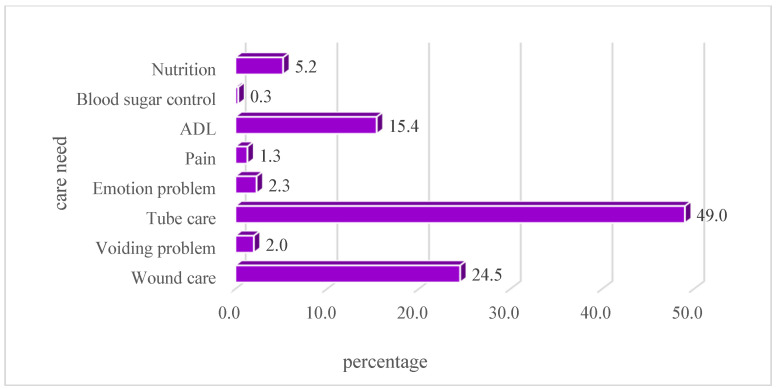
Percentage of care needs as assessed by ED nurse.

**Table 1 healthcare-11-01726-t001:** Characteristics of participants (*n* = 1187).

Variable	*n*	%	M (SD)
Gender			
Male	598	50.4	
Female	589	49.6	
Age (Range: 65–103)			77.44 (7.88)
65–74	468	39.46	
75–84	458	38.62	
>85	260	21.92	
Health Problems			
Non-Trauma	960	80.9	
Trauma	227	19.1	
Return Visit Time			
Within 24 h	500	42.2	
24–48 h	394	33.2	
48–72 h	292	24.6	
Triage			
Emergency (level 2 of TTAS)	230	18.39	
Urgent (level 3 of TTAS)	956	80.61	
Reasons for Return Visits			
Pertaining to Previous Disease/Treatment	572	48.2	
New Medical Condition	615	51.8	
Revisit Outcome			
Discharge From ED	785	66.3	
Admission	346	29.2	
Transfer to Another Hospital	27	2.3	
AAD	19	1.6	
Expire	7	0.6	
TRST			
Cognitive Impairment	166	14.0	
Difficulty Walking/Transferring	337	28.4	
Falls Within One Month	128	10.8	
Polypharmacy	577	48.7	
Previous ED Visited Within 30 Days	592	49.9	
Hospitalization Within 120 Days	289	24.3	
Live Alone	8	0.7	
Without a Caregiver	825	69.5	
Care Needs as Assessed by Nurses	291	24.5	
History of Chronic Diseases	1114	93.9	

**Table 2 healthcare-11-01726-t002:** Comparison of demographic variable and TRST among different time points (*n* = 1187).

Variable	<24 h	OR (95%CI)/*p*	24–48 h	OR (95%CI)/*p*	48–72 h	OR (95%CI)/*p*
	Revisit%		Revisit%		Revisit%	
Age						
65–74	44.4	1	32.9	1	22.6	1
75–84	42.1	1.10 (0.85–1.43) *p* = 0.479	33.4	0.98 (0.74–1.29)/*p* = 0.871	24.5	0.90 (0.67–1.23)/*p* = 0.517
≥85	37.9	1.31 (0.96–1.78) *p* = 0.088	33.3	0.98 (0.71–1.35)/*p* = 0.906	28.2	0.74 (0.52–1.05) *p* = 0.087
Reasons for Return Visits						
Previous Problem	53.8	2.57 (2.06–3.21) *p* < 0.0001	29.5	0.68 (0.54–0.85)/*p* = 0.0008	16.5	0.447 (0.345–0.58)/*p* < 0.0001
New Medical Conditions	31.2		38.1		30.7	
Revisit Destination						
Admission	44.7	1.11 (0.88–1.42) *p* = 0.76	32.0	0.898 (0.7–1.16)/*p* = 0.411	23.3	0.99 (0.75–1.31) *p* = 0.938
Discharge	42.0		34.3		23.5	
Revisit Destination						
Admission	44.7	1.11 (0.88–1.42) *p* = 0.76	32.0	0.898 (0.7–1.16)/*p* = 0.411	23.3	0.99 (0.75–1.31) *p* = 0.938
Discharge	42.0		34.3		23.5	
TRST						
Cognitive Impairment						
Yes	46.4	1.22 (0.88–1.7) *p* = 0.234	24.7	0.62 (0.43–0.91)/*p =* 0.013	28.9	1.29 (0.9–1.86)/*p =* 0.167
No	41.5		34.5		23.9	
Difficulty Walking/Transferring						
Yes	38.3	0.8 (0.62–1.03) *p* = 0.088	31.5	0.9 (0.69–1.18)/*p =* 0.44	30.3	1.51 (1.13–1.99) *p* = 0.0046
No	43.7		33.8		22.4	
Falls Within One Month						
Yes	37.5	0.81 (0.55–1.18) *p* = 0.262	36.7	1.19 (0.81–1.74)/*p* = 0.37	25.0	1.02 (0.67–1.57) *p* = 0.911
No	42.7		32.8		24.6	
Polypharmacy						
Yes	43.7	1.13 (0.896–1.42)/*p* = 0.303	30.5	0.79 (0.62–1.01)/*p* = 0.06	25.8	1.13 (0.87–1.48) *p* = 0.35
No	40.7		35.6		23.5	
Previous ED Visited Within 30 Days						
Yes	41.2	0.93 (0.74–1.17) *p* = 0.511	33.4	1.03 (0.81–1.31)/*p* = 0.821	25.2	1.06 (0.81–1.38) *p* = 0.661
No	43.1		32.8		24.1	
Hospitalization Within 120 Days						
Yes	42.2	1.005 (0.77–1.31)/*p* = 0.971	36.3	1.2 (0.81–1.31)/*p* = 0.821	21.1	1.06 (0.81–1.38) *p* = 0.661
No	42.1		32.2		25.7	
Without a Caregiver						
Yes	41.9	0.98 (0.76–1.25)/*p* = 0.846	33.1	0.97 (0.74–1.26)/*p* = 0.805	25.0	1.07 (0.8–1.43) *p* = 0.655
No	42.5		33.7		23.8	
Care Needs as Assessed by Nurses						
Yes	47.4	1.33 (1.02–1.74) *p* = 0.0354	30.2	0.84 (0.63–1.11)/*p* = 0.218	22.3	0.85 (0.62–1.16) *p* = 0.302
No	40.4		34.2		25.3	

**Table 3 healthcare-11-01726-t003:** Predictors of return visits to the emergency department.

Variable	Revisited Within 24 h					
	β	SEM	Wald	*p*	Exp B	95% CI
Constant	−0.67	0.44	2.26	0.13	0.51	
Age						
65–74	0.27	0.16	2.70	0.100	1.31	0.95–1.81
75–84	0.18	0.16	1.26	0.262	1.20	0.87–1.66
≥85	*					
TRST						
Cognitive Impairment	−0.42	0.22	3.64	0.056	0.65	0.42–1.01
Difficulty Walking/Transferring	0.56	0.18	9.79	0.002	1.76	1.23–2.50
Falls Within One Month	0.15	0.20	0.53	0.466	1.16	0.78–1.72
Polypharmacy	−0.23	0.13	3.04	0.081	0.80	0.62–1.03
Previous ED Visited Within 30 Days	0.12	0.14	0.84	0.361	1.13	0.87–1.47
A Hospitalization Within 120 Days	−0.03	0.15	0.03	0.859	0.97	0.72–1.31
Without a Caregiver	−0.03	0.14	0.03	0.860	0.98	0.74–1.29
Care Needs as Assessed by Nurses	−0.42	0.17	6.38	0.012	0.66	0.48–0.91
History of Chronic Diseases	0.51	0.26	3.77	0.052	1.67	0.99–2.79
Revisited Within 24–48 h						
Constant	1.31	0.47	7.56	0.006	3.69	
Age						
65–74	−0.10	0.17	0.37	0.545	0.90	0.64–1.26
75–84	−0.05	0.17	0.10	0.750	0.95	0.68–1.32
≥85	*					
TRST						
Cognitive Impairment	0.53	0.24	4.90	0.027	1.70	1.06–2.72
Movement Disability	−0.11	0.18	.35	0.555	0.90	0.63–1.28
Falls Within One Month	−0.23	0.20	1.32	0.250	0.79	0.53–1.18
Polypharmacy	0.33	0.14	5.95	0.015	1.40	1.07–1.83
Previous ED Visited Within 30 Days	−0.01	0.14	.00	0.968	0.99	0.75–1.31
A Hospitalization Within 120 Days	−0.30	0.16	3.66	0.056	0.74	0.54–1.01
Without a Caregiver	0.01	0.15	.01	0.930	1.01	0.76–1.36
Care Needs as Assessed by Nurses	0.05	0.17	.09	0.765	1.05	0.75–1.48
History of Chronic Diseases	−0.33	0.29	1.36	0.243	0.72	0.41–1.25
Revisited Within 48–72 h						
Constant	1.69	0.52	10.62	0.001	5.44	
Age						
65–74	−0.20	0.18	1.20	0.273	0.82	0.57–1.17
75–84	−0.14	0.18	0.60	0.439	0.87	0.61–1.24
≥85	*					
TRST						
Cognitive Impairment	−0.08	0.24	0.12	0.731	0.92	0.57–1.48
Difficulty Walking/Transferring	−0.57	0.19	9.01	0.003	0.57	0.39–0.82
Falls Within One Month	0.14	0.23	0.41	0.521	1.16	0.74–1.80
Polypharmacy	−0.12	0.15	0.62	0.432	0.89	0.67–1.19
Previous ED Visited Within 30 Days	−0.14	0.15	0.83	0.364	0.87	0.64–1.18
A Hospitalization Within 120 Days	0.42	0.18	5.48	0.019	1.52	1.07–2.16
Without a Caregiver	0.04	0.17	0.06	0.809	1.04	0.75–1.44
Care Needs as Assessed by Nurses	0.48	0.20	6.04	0.014	1.62	1.10–2.37
History of Chronic Diseases	−0.30	0.33	0.85	0.356	0.74	0.39–1.41

* reference group.

## Data Availability

All data have been illustrated in the manuscript.

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
