# Peer review of "Factors Associated with Return Visits by Elders within 72 Hours of Discharge from the Emergency Department"

_healthcare, 2023, doi:10.3390/healthcare11121726_

Round 1

Reviewer 1 Report

Overall review: 

I am glad I got to read this script. It is urgent research. The article is of great importance because the care of elderly people who need emergency medical care is urgent. In large parts of the world, there is an increased number of elderly people who have and will need healthcare. Finding reasons for return visits is helpful for health care organizations. By predicting the risk factors in return visits, more precise measures are possible. The difficulty, however, as with all evidence, is to actually implement it.

Areas of the strength of this paper:

Introduction:

·       The chosen research problem is problematized, motivated and defined based on the current state of the art. But see below under the heading References to specify the choice.

·       The introduction is predominantly clear and related to the problem area.

The purpose is relevant, clearly formulated and limited to the problem area.

Materials and Methods: 

·       The choice of method is described and it is clear what the selection is.

·       Data analysis is well described and credible.

Results:

·       The results is structured and based on a correct and systematic analysis. Characteristics of return visits by elders is well described.

·       The tables and figure are helpful.

Discussion: 

Main results are discussed with a reflective and critical approach in relation to the current state of research. Strengths and weaknesses in relation to the result are discussed with a critical approach.

Areas of theweakness of this paper: 

Introduction:

·       In the title the first word should be in plural because there is more than one factor in the result.Factors Associated with Return Visits by Elders within 72 Hours of Discharge from the Emergency Department”

·       Rows 41-42. Do you mean that the frequency of reported return visits in the three studies that you refer to varied the frequency of return visits between 0.92% and 27%? Were these patients different ages and different disease states? I ask because your study is about return Visits by Elders.

Materials and Methods: 

·       The selected instrument is well described and it is stated that it is modified. However, it would be if it were made clear about how much it is used, its validity and reliability.

Results:

The text under the heading Return visit reason and time is a bit difficult to understand. If possible, try to clarify it. However, the table is helpful.

References: 

The references are mostly very relevant and recent, one is from 2013.

·       Row 32 The first sentence about global indicator of monitor-32 ing the quality of emergency care.lacks reference.

·       Also, somwhere in the manusscript it would be of interest to raise something about the fact that in different parts of the world there are different age pyramids, for example on line 46 after where you write about "The increasing aging population has a significant impact on the demand for ED services and presents a global challenge to emergency care."

·       Reference number 1 in the reference list is a qualitative study and cannot answer whether the ED return visit rate among adults aged 65 years and above is higher than other populations.

Author Response

Response to Reviewer Comments

Thank you very much for the opportunity to respond to the reviewer’s comments. Authors were delighted with the feedback from the reviewers and have addressed the suggested textual changes through the additions and corrections recorded in the manuscript.

Reviewer 1

Reply

I am glad I got to read this script. It is urgent research. The article is of great importance because the care of elderly people who need emergency medical care is urgent. In large parts of the world, there is an increased number of elderly people who have and will need healthcare. Finding reasons for return visits is helpful for health care organizations.

By predicting the risk factors in return visits, more precise measures are possible. The difficulty, however, as with all evidence, is to actually implement it.

Thank you reviewer 1.

Introduction:

In the title the first word should be in plural because there is more than one factor in the result. “Factors Associated with Return Visits by Elders within 72 Hours of Discharge from the Emergency Department”

Edited the title to plural.

        Rows 41-42. Do you mean that the frequency of reported return visits in the three studies that you refer to varied the frequency of return visits between 0.92% and 27%? Were these patients different ages and different disease states? I ask because your study is about return Visits by Elders.

Edited in line 43-45.

Materials and Methods: 

The selected instrument is well described and it is stated that it is modified. However, it would be if it were made clear about how much it is used, its validity and reliability.

Added more information in the Methods section in line 98-117.

Results:

The text under the heading Return visit reason and time is a bit difficult to understand. If possible, try to clarify it. However, the table is helpful.

Edited in line 163-164.

References: 

The references are mostly very relevant and recent, one is from 2013.

Deleted the reference in 2013.

Row 32 The first sentence about global indicator of monitoring the quality of emergency care. lacks reference.

Cited reference in line 33.

Also, somewhere in the manuscript it would be of interest to raise something about the fact that in different parts of the world there are different age pyramids, for example on line 46 after where you write about "The increasing aging population has a significant impact on the demand for ED services and presents a global challenge to emergency care."

Added more information in line 46-55.

Reference number 1 in the reference list is a qualitative study and cannot answer whether the ED return visit rate among adults aged 65 years and above is higher than other populations.

Amended

Reviewer 2 Report

Dear authors and editor,

Thank you for the opportunity to review the manuscript entitled “Factor Associated with Return Visits by Elders within 72 Hours”. This topic is highly relevant for both patients, clinicians and the society as a whole. The study’s results offer new valuable knowledge when caring for elders in the ED. The aim of the study was to determine the factors associated with return visits to the ED by elders within 72 hours after discharge and to predict the risk factors in return visits within different amounts of time after discharge. A retrospective medical record review was carried out. The setting was a regional hospital in Taiwan. The elders risk factors were identified using the Triage Risk Screening Tool (TRST). Elders included in the study made more frequent return visits to the ED compared to the overall adult population. Elders with tubing and wound care needs had higher risks of return visits to ED.

Abstract:

Discharge from what (ED? hospital ward? Combination of both?) – please clarify in the abstract and in the main text.

Overall:

Is there any definition of early risk factors? Please explain the difference between early risk factors and risk factors as you use early risk factors in the study.

Aim:

Reflect upon if the word early should be included in the aim of the study (early risk factors). The authors themselves point out the study’s novelty by highlighting early risk factors when comparing with previous studies that have reported risk factors.

Materials and methods:

Data collection is missing – please report on this subject in detail. Please expand on how data were retrieved and managed? How many registers were used? If more than one – how was the data merged? Was any validity test performed after merging the data? Was data manually or automatically extracted from the included medical records? Who extracted the data, more than one person? How was the inter reader reliability managed and tested for? Were any validity checks through a limited random sample of the included medical records and the extracted data considered or performed?

If a patient had more than one return visit within 72-hours, how this managed in the data handling process? Were return visits to return visits considered eligible for inclusion? How was the index even defined?

Please include a data inclusion flow chart visualising the inclusion and exclusion process of patients.

Data analysis:

Have you had any thoughts on using additional statistical calculations/modelling? Such as calculating the incidence rate ratio.

Results:

Row 111: The Taiwan triage and acuity scale are mentioned, please describe what level 3 indicates (high, intermediate or low priority).

Row 153: Please review the sentence where elders with cognitive impairment are described as having higher rate of return visits within 24-48 hours – compared to who? The OR presented both in table 2 and in this sentence is < 1 hence indicating a protective effect. Both the rate and the percentage of patients with cognitive impairment who have return visits are lower than patients without cognitive impairment.

Row 178: Elders with walking difficulties had higher risk of return visits within 24 hours but not within 24-48 – how come you think?

Row 192: Follow-up outcomes of the return visits are mentioned shortly. If you have data on these outcomes the study would be strengthened from presenting these.

Discussion:

Overall, a well written section in which the authors contextualize, compare and interpret the study’s result. However, new data are presented in the discussion (row 215), data that have not been presented in the result section. The discussion section should not include data, especially not data that the reader has not seen before. Furthermore, a paragraph about future research is missing and would strengthen the discussion. Based on the study´s results, what future research is needed?

Conclusion:

Sufficient and supported by the findings.

Table 2:

Difficult to get a good overview and thus understanding of the table and its content due to its current layout.
Clarify which variables were used as references in the logistic regression analysis. It is only obvious for the age where the variable 65-74 were used a reference.

Author Response

Response to Reviewer Comments

Thank you very much for the opportunity to respond to the reviewer’s comments. Authors were delighted with the feedback from the reviewers and have addressed the suggested textual changes through the additions and corrections recorded in the manuscript.

Reviewer 2

Abstract:

Discharge from what (ED? hospital ward? Combination of both?) – please clarify in the abstract and in the main text. 

Thank you reviewer 2. Edited in line 20.

Overall:

Is there any definition of early risk factors? Please explain the difference between early risk factors and risk factors as you use early risk factors in the study.

Deleted the word of “early”. We would like to identify risk factors for return visits within different amounts of time following discharge Sorry to be confused. 

Aim:

Reflect upon if the word early should be included in the aim of the study (early risk factors). The authors themselves point out the study’s novelty by highlighting early risk factors when comparing with previous studies that have reported risk factors.

Deleted the word of “early”. We would like to identify risk factors for return visits within different amounts of time following discharge Sorry to be confused.

Materials and methods:

Data collection is missing – please report on this subject in detail.

Please expand on how data were retrieved and managed?

How many registers were used? If more than one – how was the data merged?

Was any validity test performed after merging the data? Was data manually or automatically extracted from the included medical records?

Who extracted the data, more than one person? How was the inter reader reliability managed and tested for?

Were any validity checks through a limited random sample of the included medical records and the extracted data considered or performed?

Added data collection and the more details how to collect data for the study in line 97-117

If a patient had more than one return visit within 72-hours, how this managed in the data handling process? Were return visits to return visits considered eligible for inclusion? How was the index even defined?

Added more information related to the inclusion and exclusion criteria of the study in line 78-81.

Please include a data inclusion flow chart visualising the inclusion and exclusion process of patients.

Added a flow chart in line 129-146.

Results:

Row 111: The Taiwan triage and acuity scale are mentioned, please describe what level 3 indicates (high, intermediate or low priority).

Added more information related to the Taiwan triage and acuity scale in line 123, 155 and table 1.

Row 153: Please review the sentence where elders with cognitive impairment are described as having higher rate of return visits within 24-48 hours – compared to who? The OR presented both in table 2 and in this sentence is < 1 hence indicating a protective effect. Both the rate and the percentage of patients with cognitive impairment who have return visits are lower than patients without cognitive impairment.

Thank you review. amended content in line 209-211.

Row 178: Elders with walking difficulties had higher risk of return visits within 24 hours but not within 24-48 – how come you think?

Added in discussion line 306-311.

Discussion:

Overall, a well written section in which the authors contextualize, compare and interpret the study’s result. However, new data are presented in the discussion (row 215), data that have not been presented in the result section. The discussion section should not include data, especially not data that the reader has not seen before.

Thank you review. The new data are presented in the discussion (row 215) has been deleted.

Furthermore, a paragraph about future research is missing and would strengthen the discussion. Based on the study´s results, what future research is needed?

Added further research in line 322-326.

Conclusion:

Sufficient and supported by the findings.

Thank you reviewer 2.

Table 2:

Difficult to get a good overview and thus understanding of the table and its content due to its current layout.
Clarify which variables were used as references in the logistic regression analysis. It is only obvious for the age where the variable 65-74 were used a reference.

Adjusted table 2 and table 3. Deleted some variables that were not used in line 202-203 and 245-246. 

Reviewer 3 Report

Dear All,

Having analysed the work entitled “Factor Associated with Return Visits by Elders within 72 Hours of Discharge from the Emergency Department” I can confirm that:

1)     The content of the article is consistent with the aim of the article.

2)     The work tackles an issue that is of particular importance from the point of view of gerontology and the organisation of care for the elderly, especially in relation to the emergency department.

3)     The work fills gaps in the already published literature.

4)  The methodology does not raise any objections. The authors should justify, however, why the year 2017 in particular was analysed. It might be also interesting for the readers to learn the rules of functioning of nurses in the emergency department in the context of preparing instructions for care.

5)     Conclusions correspond to the collected research material.

6)     The research literature was correctly selected, however, the authors might want to consider refraining from referencing works published in 1987, 2001 and 2003.

7)     The presented tables are legible and do not raise any doubts.

8)     The heading “Discussion" appears twice in the text. It would be wise to unify the writing style, mainly in the Discussion section.

Author Response

Response to Reviewer Comments

Thank you very much for the opportunity to respond to the reviewer’s comments. Authors were delighted with the feedback from the reviewers and have addressed the suggested textual changes through the additions and corrections recorded in the manuscript.

Reviewer 3

The content of the article is consistent with the aim of the article.    

Thank you reviewer 3.

 The work tackles an issue that is of particular importance from the point of view of gerontology and the organisation of care for the elderly, especially in relation to the emergency department.

Thank you reviewer 3.

The work fills gaps in the already published literature.

Thank you reviewer 3.

The methodology does not raise any objections. The authors should justify, however, why the year 2017 in particular was analysed. It might be also interesting for the readers to learn the rules of functioning of nurses in the emergency department in the context of preparing instructions for care.

Added the reasons in line 72-75.

 Conclusions correspond to the collected research material.

Thank you reviewer 3.

The research literature was correctly selected, however, the authors might want to consider refraining from referencing works published in 1987, 2001 and 2003. 

Deleted old references and updated.

The presented tables are legible and do not raise any doubts.

Thank you reviewer 3.

The heading “Discussion" appears twice in the text. It would be wise to unify the writing style, mainly in the Discussion section.

deleted the twice text.

Reviewer 4 Report

I would like to thank the authors for their work. This is an interesting paper, which aims to determine the factors associated with return visits to the ED by elders within 72 hours after discharge and to predict the risk factors in return visits within different amounts of time after discharge.

The background is robust, but I suggest to deepen the sentence "The ED return visit rate among adults aged 65 years and above is higher than other populations" (page 1, lines 33-34): Why is this rate higher? Are there reasons cited in the literature?

The methodology is consistent with the research question; I would only ask to specify whether, despite the large sample, the normality of the distribution was tested.

The results are well stated and clear to the reader.

The discussion is not speculative, and the conclusions are consistent with the study.

The limitations are explicit.

Author Response

Response to Reviewer Comments

Thank you very much for the opportunity to respond to the reviewer’s comments. Authors were delighted with the feedback from the reviewers and have addressed the suggested textual changes through the additions and corrections recorded in the manuscript.

Reviewer 4

The background is robust, but I suggest to deepen the sentence "The ED return visit rate among adults aged 65 years and above is higher than other populations" (page 1, lines 33-34): Why is this rate higher? Are there reasons cited in the literature?

Added more information in line 46-55.

The methodology is consistent with the research question; I would only ask to specify whether, despite the large sample, the normality of the distribution was tested.

Yes. Added in line 119-120.

The results are well stated and clear to the reader.

Thank you Reviewer 4.

The discussion is not speculative, and the conclusions are consistent with the study.

Thank you Reviewer 4.

The limitations are explicit.

Thank you Reviewer 4.